# Assessing the Cost-Effectiveness of Interventions That Simultaneously Prevent High Body Mass Index and Eating Disorders

**DOI:** 10.3390/nu12082313

**Published:** 2020-07-31

**Authors:** Long Khanh-Dao Le, Phillipa Hay, Jaithri Ananthapavan, Yong Yi Lee, Cathrine Mihalopoulos

**Affiliations:** 1Deakin Health Economics, Institute for Health Transformation, School of Health and Social Development, Deakin University, Burwood, Victoria 3125, Australia; jaithri.ananthapavan@deakin.edu.au (J.A.); yongyi.lee@deakin.edu.au (Y.Y.L.); cathy.mihalopoulos@deakin.edu.au (C.M.); 2Translational Health Research Institute (THRI), School of Medicine, Western Sydney University, Locked Bag 1797, Penrith, NSW 2751, Australia; p.hay@westernsydney.edu.au; 3Camden and Campbelltown Hospital, SWSLHD, Campbelltown, NSW 2560, Australia; 4Global Obesity Centre, Institute for Health Transformation, School of Health and Social Development, Deakin University, Burwood, Victoria 3125, Australia; 5School of Public Health, The University of Queensland, Herston, QLD 4006, Australia; 6Policy and Epidemiology Group, Queensland Centre for Mental Health Research, Wacol, QLD 4076, Australia

**Keywords:** eating disorders, obesity, high BMI, prevention, economic evaluation, cost-effectiveness

## Abstract

Eating disorders (ED) are among the top three most common debilitating illnesses in adolescent females, while high Body Mass Index (BMI) is one of the five leading modifiable risk factors for preventable disease burden. The high prevalence of eating and weight-related problems in adolescence is of great concern, particularly since this is a period of rapid growth and development. Here, we comment on the current evidence for the prevention of EDs and high BMI and the importance of assessing the cost-effectiveness of interventions that integrate the prevention of EDs and high BMI in this population. There is evidence that there are effective interventions targeted at children, adolescents and young adults that can reduce the prevalence of risk factors associated with the development of EDs and high BMI concurrently. However, optimal decision-making for the health of younger generations involves considering the value for money of these effective interventions. Further research investigating the cost-effectiveness of potent and sustainable integrated preventive interventions for EDs and high BMI will provide decision makers with the necessary information to inform investment choices.

## 1. Introduction

The high prevalence of eating and weight-related problems during childhood and adolescence is of great concern, particularly since these ages are periods of rapid growth and development. Disordered eating patterns are a common precursor to eating disorders (EDs). EDs are among the top three most common chronic illnesses in adolescent females [1]. The public health and cost impacts of high body mass index (BMI) are well known [2,3]. There is also evidence that the risk factors for developing EDs and high BMI can be prevented using the same interventions in childhood, adolescence and even young adulthood. Unfortunately, there has been no comprehensive evaluation of such preventive strategies on the combined outcomes of EDs and high BMI, preventing optimal decision-making for the health of our younger generations. Robust economic evaluation of such interventions is imperative to provide decision makers with the necessary information to inform investments as well as to develop clinical guidelines for communicating with health professionals. 

In this commentary we outline the rationale for the “Assessing Cost-Effectiveness (ACE) of the Prevention of High Body Mass Index and Eating Disorders in Australia (ACE-HiBED)” project, a world-first study that uniquely brings together EDs and high BMI in a way that acknowledges shared upstream determinants. 

## 2. Eating Disorders and High BMI Are Significant Problems in Children and Adolescents

Approximately one in three children and adolescents (aged 5–17 years) globally have disordered eating [4] and a further one in five (aged 5–19 years) are overweight or obese (defined as having BMI z-scores over the 95th percentile for age and sex) [5]. EDs typically begin in adolescence, although they are increasingly being recognised in children as young as 5 years [6,7]. EDs have been associated with functional impairment, distress, physical and psychiatric morbidity and increased mortality [8,9,10,11]. Furthermore, the costs of treating EDs once they develop are high, as current evidence-based treatments are very resource-intensive [12,13]. Annual health care costs for EDs have been estimated to be A$88 million [14] across all age groups in Australia. High BMI is one of the five leading modifiable risk factors for preventable disease burden according to the global burden of disease studies [2]. Elevated BMI in children costs Australian society over A$43 million annually [3] and A$8.6 billion across the whole population [5]. 

Recent research indicates that body image-related symptoms such as fear of weight gain and overvaluation of weight were prevalent amongst children and adolescents aged 5–12 years, independent of weight status. The high prevalence of eating and weight-related problems during the ages of 5–19 years is of great concern, as this is a period of rapid growth and development. Children and adolescents with high BMI have also demonstrated an increased risk of ED risk factors [15] and a higher risk of developing EDs compared to those with healthy weight [16]. Compensatory behavioural symptoms such as vomiting and excessive exercise have been reported to be common among adolescents with high BMI [17]. Whilst EDs and high BMI have traditionally been conceptualised as separate problems, with independent trajectories and methods of treatment and prevention [18], recent studies have found that they share multiple psychological and environmental risk factors such as dieting, body dissatisfaction, media exposure, perfectionism, trauma, weight teasing by family and friends and internalisation of the cultural beauty ideal [19,20]. This supports a close aetiological association [16]. Further, adolescent girls with restrictive dieting and unhealthy weight control behaviours have a two-fold likelihood of high BMI compared to females not engaging in these behaviours [21]. In addition, these disorders also share upstream determinants. Children and adolescents are exposed to food and physical environments that encourage disordered eating and low levels of physical activity contributing to both EDs and high BMI. Children and adolescents are heavily exposed to marketing of discretionary foods [22] and at the same time exposed to potentially harmful and misleading marketing from the dieting industry [23]. There is considerable evidence from various countries that the increasing rate of discretionary, ultra-processed food consumption is associated with the rapid increase of obesity [24,25,26]. Highly-processed foods, containing refined sugars and added fats, have also been identified as being associated with loss of control eating [27,28] and potentially triggering binge eating [29,30]. 

## 3. Prevention of Eating Disorders and High BMI—What We Know so Far

There is inconsistent evidence as to whether prevention or treatment of high BMI is associated with increased risk of EDs and vice versa. A large systematic review conducted by the National Eating Disorders collaboration investigated the impact of obesity treatment interventions on EDs and found no evidence of impact on EDs [20]. However, there is limited evidence from a recent smaller systematic review that exposure to both mass media and anti-obesity public health messages can have adverse effects on audiences and could intensify thin internalisation and drive for thinness [31]. Despite the intersection between EDs and elevated BMI, a key limitation of obesity prevention and weight-loss treatment intervention studies is the limited use of validated tools to measure EDs, and until there is more consistent measurement of EDs, the impact on this important health outcome will remain unclear [20,31]. Although ED prevention studies tend to measure BMI more frequently [32], the measurement of BMI alone may be inadequate as studies require long-term follow up in order to demonstrate impact on BMI. Therefore, studies aimed at preventing EDs should also measure intermediate outcomes such as diet quality and physical activity levels [33]. 

Given the clear link between EDs and high BMI, there is emerging evidence that interventions can be effective in preventing both disordered eating behaviours and unhealthy weight gain [32]. A meta-analysis by Le et al. (2017) [32] found that healthy weight interventions significantly reduced both ED symptoms and BMI. For example, the Healthy Weight program is a three-hour, school-based intervention that aimed to prevent ED. When compared to assessment-only controls, this program not only reduced ED onset by 61% but also prevented cases of obesity onset by 55% at 3-year follow up [34]. Positive results on ED outcomes have also been reported in some obesity prevention studies. For example, the Planet Health Program (an obesity prevention intervention) prevented over half of expected new cases of disordered weight-control behaviour among females [35]. Future research in this area should consider the short- and longer-term impacts across both ED and elevated BMI in intervention development, monitoring and evaluation. 

While ED and high BMI prevention programs have the capacity to reduce the onset of two important and interrelated public health problems, it is unclear whether the routine roll-out of such interventions will result in an affordable and efficient use of resources from a government perspective. A true assessment of the value for money of these interventions requires the assessment of the full range of benefits and risks alongside the potential investment required. 

## 4. Integrating Prevention of Eating Disorders and High BMI: A Question of Value for Money

Economic evaluation is an analytic tool that assesses the value for money of interventions and can therefore assist decision makers in prioritising interventions for implementation. Economic evaluations conducted for specific decision contexts can inform which interventions are not only good value for money but also affordable and implementable in the local context [36]; which interventions are likely to be good value for money but need further evaluation; and which interventions do not represent good value for money and therefore should not be implemented. 

To our knowledge, interventions that integrate the simultaneous prevention of both high BMI and EDs have not been considered in any previous cost-effectiveness or priority-setting studies. In Australia, there have been several priority-setting studies that have separately evaluated the economic credentials of interventions that either prevent high BMI or EDs. These studies have utilised the Assessing Cost-Effectiveness (ACE) methodology and include: two obesity-related priority-setting studies (ACE-Obesity and ACE-Obesity Policy) [37,38]; one mental health study (ACE-Mental Health) [39]; and one prevention-related study (ACE-Prevention) which separately evaluated preventive interventions for obesity and mental health, but not concurrently [40]. It is important to note that previous evaluations might over- or under-estimate the cost-effectiveness of preventive interventions, as the impact of these various intervention on both ED and high BMI has not been previously investigated. 

Furthermore, the current methods used in the ACE studies to assess cost-effectiveness are likely to underestimate the impact of ED prevention as they used outcome measures not sensitive to changes in ED risk factors. The disability weights (used to calculate disability-adjusted life years (DALYs) or health-adjusted life years (HALYs)) in previous ACE studies that were assigned to health states do not adequately account for the gradient of illness severity experienced by people with EDs. Although there are different disability weights for the various levels of illness severity experienced by people with depression and anxiety (mild, moderate and severe), there is only a single disability weight for the two main types of EDs (i.e., one for anorexia nervosa and one for bulimia nervosa). Therefore, any reduction in illness severity (e.g., change from a more serious ED to a mild/subsyndromal ED) would not be adequately captured by the existing ED disability weights [41]. This may result in interventions that reduce illness severity but do not result in complete remission appearing to be less cost-effective than they actually are. In order to integrate the evaluation of interventions that simultaneously impact both ED and high BMI, methods and economic models for incorporating health state valuations that are sensitive to illness severity are required.

Lastly, broader societal impacts outside the health sector, including productivity and educational impacts, have not been captured in previous cost-effectiveness studies, especially ED prevention [42]. Existing ACE studies have been undertaken using a health sector perspective, where all costs and benefits relevant to government departments of health, and health care-related costs and benefits pertaining to the individual and other third-party payers (e.g., private health insurance) were incorporated. An exception to this is ACE-Obesity Policy which attempted to measure costs incurred beyond the health care sector (e.g., time and travel costs and industry revenue) but excluded impacts on productivity and educational attainment. 

## 5. Challenges and Complexities in Translating Research Outcomes into Health Policy and/or Practice within the Australian Context

The organisation, delivery and funding of the Australian health care system is complex. There is no one unifying framework for evaluating the cost-effectiveness of different health care interventions. Pharmaceuticals that seek to be listed on the Pharmaceutical Benefits Scheme (PBS) and medical technologies (or allied health interventions) that seek to be listed on the Medicare Benefits Schedule (MBS) must undergo a process of evaluation of their quality, safety, efficacy and cost-effectiveness. This is done through a formal health technology assessment (HTA) framework whereby such products are evaluated via the Pharmaceutical Benefits and Medical Services Advisory Committees (PBAC/MSAC) before recommendations for funding are made. In both these settings, cost–utility analyses (CUA) using quality-adjusted life years (QALYs) are core to decision-making. 

These HTA frameworks are not routinely used outside of the MBS/PBS schemes. Many mental health and obesity prevention and treatment interventions are not funded via the MBS/PBS and therefore are not subjected to formal HTA requirements. Furthermore, many effective interventions require financial support and implementation from sectors outside of health (e.g., school-based interventions typically involve the education sector). In these instances, it is unclear how economic evidence is currently used in funding decisions. This is particularly a problem for preventive health interventions which are typically complex and multisectoral. In these cases, investment decisions are usually based on direct appeals to Treasury departments, often with submissions that include “business case” arguments. Departments of Treasury tend to recommend cost–benefit analyses (CBA) undertaken from a societal perspective where all costs and benefits are valued in monetary terms [43]. The decision criterion of CBA is simple—if monetised benefits outweigh the costs of the intervention, then it is “worth” doing, with the net present value guiding ranking of multiple worthwhile projects. There are, however, well-known difficulties in attributing monetary valuations to health and mental health benefits [44]. Thus, these business case arguments adopt narrower foci and are often termed “return on investment” (ROI) analyses. These primarily consider the costs of implementing interventions compared with cost savings, such as productivity improvements or downstream health care savings. It has been argued that such frameworks are easier to understand than CUA for decision-making [45,46]. However, an intervention may appear cost-effective using a CUA framework, but not when using a ROI framework (when health benefits are not explicitly valued). 

Therefore, it is important that policy makers are aware of such differences when using these frameworks to make investment decisions on the use of limited societal resources. In addition to capturing outcomes that are relevant to other sectors, for example, educational outcomes, in order to better inform decision makers of the value for money of prevention interventions, it is imperative that researchers better understand multi-sectoral decision-making processes and the role of economic evidence in the process.

## 6. Conclusions

There is evidence of effective interventions in childhood, adolescence and young adulthood that simultaneously address risk factors for developing both EDs and high BMI. There is currently no comprehensive evaluation of such preventive strategies on the combined outcomes of EDs and high BMI, thus hindering optimal decision-making for the health of young people in Australia. The ACE-HiBED project uniquely considers the economic credentials of preventive interventions for two high burden public health issues within one priority-setting study that shifts the current paradigms where separate cost-effectiveness analyses have been conducted. Robust economic evaluation of such interventions, using various economic evaluation frameworks, is imperative to provide decision makers with the necessary information to inform investment decisions as well as clinical guidelines.

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
