# Peer review of "Assessing the Cost-Effectiveness of Interventions That Simultaneously Prevent High Body Mass Index and Eating Disorders"

_nutrients, 2020, doi:10.3390/nu12082313_

Round 1

Reviewer 1 Report

In this commentary article, the authors provide an argument for conducting the ACE-HiBED cost-effectiveness analysis. Efforts to inform decision makers about the costs of interventions are of high value for increasing access to care, making the proposed ACE-HiBED project an important research endeavor. 

However, the article itself was difficult to follow, and the authors might consider substantially reorganizing the material. It is very unclear the point of the article until the final conclusion paragraph (Section 6). In fact, it seems misleading to call this a case study, and label Section 5 as the case study, without actually presenting any data on or from the case study. The authors might consider "flipping" the order of their article by presenting the details of the concluding paragraph at the beginning of the paper, and then providing their justification for the ways in which ACE-HiBED represents substantive improvements over past economic analyses, and why an analysis addressing the combination of eating disorder and weight gain prevention interventions are warranted. As an aside, without a clear direction for the paper, Section 2 felt particularly disjointed. If the authors elect to make the suggested revisions to the order of the paper, they might consider whether this section is necessary (of note, the brief final paragraph at the end of Section 2 likely is sufficient with some references added). 

As mentioned above, it seems misleading to refer to this paper as a case study without presenting the results of the case study itself. Perhaps this paper is meant to serve as an introductory/protocol-like overview for the ACE-HiBED project, but that may not be an accurate assessment. In both the title and manuscript, the authors are encouraged to be clear this is a project that is currently underway (assuming this interpretation is correct, although results from the project are not presented here).

A few additional changes are suggested: 

  • Section 1: The authors are encouraged to use people-first language (i.e., the terms "overweight and obesity") rather than "as overweight or obese."
  • Section 1: The authors describe that children with high BMI are at increased risk of ED risk factors and developing EDs - it might be helpful to indicate compared to whom.
  • Section 3: There may be extra/missing words in the first sentence. 
  • Section 3: It might be helpful for the unfamiliar reader to define "best buy" intervention. The Kazdin, Fitzsimmons-Craft, & Wilfley 2017 paper could be a helpful reference (PMCID: PMC6169314).

Reviewer 2 Report

This is a nicely written commentary about the important topic:

Assessing the Cost-Effectiveness of Interventions
That Simultaneously Prevent High Body Mass Index
and Eating Disorders

My main concern is that the authors only included one half a sentence on the importance of public health interventions, such as addressing the food environment, and did not include any other reference to other important topics, such as the importance of physical environments, which encourage active transport and physical activity. There is considerable evidence from various countries, including Australia, that the increasing rate Ultraprocessed food consumption is associated with rapid increase of obesity and that this highly influenced by misleading advertising. Similarly, adverts from the dieting industry can be harmful for young people with eating disorders. Without addressing this, preventative interventions aimed at individuals are unlikely to be successful. Therefore, in my opinion, the paper is not balanced and misses a large body of relevant literature. It could be improved by including these topics.

Round 2

Reviewer 1 Report

The authors were very responsive to the recommendations of the previous review, and the paper is much clearer. The presented research agenda represents an important project for eating disorders and obesity prevention.